# Inhaled nebulized glatiramer acetate against Gram-negative bacteria is not associated with adverse pulmonary reactions in healthy, young adult female pigs

**Sandra M. Skovdal**[1,2,3], **Stig Hill Christiansen**[4¤a], **Karen Singers Johansen**[5], **Ole Viborg**[6], **Niels Henrik Bruun**[7], **Søren Jensen-Fangel**[2], **Ida Elisabeth Holm**[8], **Thomas Vorup-Jensen**[4]*, **Eskild Petersen**[1,2¤b]

1 Department of Clinical Medicine, Faculty of Health, Aarhus University, Aarhus, Denmark, 2 Department of Infectious Diseases, Aarhus University Hospital, Skejby, Denmark, 3 Interdisciplinary Nanoscience Center (iNANO), Faculty of Science and Technology, Aarhus University, Aarhus, Denmark, 4 Biophysical Immunology Laboratory, Department of Biomedicine, Faculty of Health, Aarhus University, Aarhus, Denmark, 5 Department of Biomedicine, Faculty of Health, Aarhus University, Aarhus, Denmark, 6 Intensive Care Unit, Department of Anesthesiology, Aarhus University Hospital, Skejby, Denmark, 7 Biostatistical Advisory Service (BIAS), Department of Public Health, Faculty of Health, Aarhus University, Aarhus, Denmark, 8 Department of Pathology, Randers Regional Hospital, Randers, Denmark

¤a Current address: Department of Molecular Medicine, Cancer and Inflammation Research, University of Southern Denmark, Odense, Denmark
¤b Current address: Department of Infectious Diseases, The Royal Hospital, Muscat, Sultanate of Oman
* vorup-jensen@biomed.au.dk

**Data Availability Statement:** All relevant data are within the manuscript or held in the public

## Abstract

The developmental speed of new antimicrobials does not meet the emergence of multidrug-resistant bacteria sufficiently. A potential shortcut is assessing the antimicrobial activity of already approved drugs. Intrudingly, the antibacterial action of glatiramer acetate (GA) has recently been discovered. GA is a well-known and safe immunomodulatory drug particular effective against Gram-negative bacteria, which disrupts biological membranes by resembling the activity of antimicrobial peptides. Thus, GA can potentially be included in treatment strategies used to combat infections caused by multidrug-resistant Gram-negatives. One potential application is chronic respiratory infections caused by *Pseudomonas aeruginosa*, however the safety of GA inhalation has never been assessed. Here, the safety of inhaling nebulized GA is evaluated in a preclinical pig model. The potential side effects, *i.e.*, broncho-constriction, respiratory tract symptoms and systemic- and local inflammation were assessed by ventilator monitoring, clinical observation, biochemistry, flowcytometry, and histopathology. No signs of bronchoconstriction assessed by increased airway peak pressure, $P_{peak}$, or decreased oxygen pressure were observed. Also, there were no signs of local inflammation in the final histopathology examination of the pulmonary tissue. As we did not observe any potential pulmonary side effects of inhaled GA, our preliminary results suggest that GA inhalation is safe and potentially can be a part of the treatment strategy targeting chronic lung infections caused by multidrug-resistant Gram-negative bacteria.

repository figshare.com: DOI 10.6084/m9.figshare.
9003164.

**Funding:** This work was supported by a grant from
the Health Research Fund of Central Denmark
Region (https://www.rm.dk/sundhed/faginfo/
forskning/region-midtjyllands-sundhedsvidens
kabelige-forskningsfond/) for studies on positively
charged, random amino acid co-polymers as
antibiotics (TECH-2014-627). TV-J was in receipt
of funding from the Danish Multiple Sclerosis
Society (https://scleroseforeningen.dk/viden-og-
nyt/om-os/about-us) for studies on the
pharmaceutical mode of action of GA (R34-A255-
B143; R62-A619-B143; R89-A1581-B143; R367-
A25613-B143; R431-A29821-B143). The funders
had no role in study design, data collection and
analysis, decision to publish, or preparation of the
manuscript.

**Competing interests:** I have read the journal's
policy and the authors of this manuscript have the
following competing interests: SHC, EP, and TVJ
are the inventors on the US patent #10166253
"Positively charged co-polymers for use as
antimicrobial agents" and an identically entitled
application to the European Patent Office
(EP3185882A1) describing the use of GA as an
antimicrobial agent. The patent is owned by Aarhus
University and Region Midtjylland, Aarhus,
Denmark, and licensed to Cycle Pharmaceuticals
Ltd, Cambridge, UK. Neither the patent nor the
mentioned application alters our adherence to all
PLOS ONE policies on sharing data and materials.
The other authors declare no competing interests.

## Introduction

The rise of multidrug-resistant Gram-negative bacteria is a major cause of concern in health-care and constitutes significant challenges in infection management [1]. The development of novel antimicrobials is slow, costly and not sufficient to meet the current challenges [2] and only few drugs are in the pipeline [3]. Thus, repurposing drugs that are already approved and well known regarding safety profiles and pharmacokinetics is highly relevant [2].

Recently, the antimicrobial effect of the immunomodulatory drug glatiramer acetate (GA) was discovered [4]. In 1996, the US Food and Drug administration (FDA) approved GA (Copaxone™, Teva Pharmaceutical Industries Ltd.), injected subcutaneously in a single daily dose of 18 mg, which is now widely prescribed as a first-line treatment of relapsing-remitting multiple sclerosis (MS) [5]. GA is generally well-tolerated, and no serious adverse events have been recorded even when administered systemically [6,7]. The clinical effects of GA are not fully elucidated but can largely be divided into activities that relate to anti-inflammatory and neuroprotective responses [8].

Recently, GA was discovered to have chemical and biophysical properties in common with the naturally occurring antimicrobial peptide LL-37, notably the ability to disrupt biological membranes [9]. Owing to the close resemblance between the two compounds, we hypothesized that the action of GA was due to interaction with cell membranes and that bacteria could be susceptible to GA. We confirmed that GA indeed disrupted bacterial cell membranes, particularly the Gram-negative *Pseudomonas aeruginosa*, *Escherichia coli* and *Acinetobacter baumannii* and to a lesser extent the Gram-positive *Staphylococcus aureus* [4]. Owing to the mechanisms of action in which the main target is the bacterial membrane combined with immunomodulatory effects, antimicrobial peptides are often effective against multidrug-resistant bacteria [10]. Furthermore, many antimicrobial peptides are effective against biofilms, which is also the case for GA [4]. Thus, GA opens up for treatment of serious infections with Gram-negatives including multidrug-resistant bacteria.

While GA has not previously been used for inhalation therapy, it is a good candidate for closer investigations for this purpose. Relevant target groups are patients chronically colonized in the airways, here among patients with fibrotic lungs or cystic fibrosis and patients in ventilation therapy with nosocomial pneumonia caused by Gram-negatives. In addition, inhalation therapy can potentially be an alternative to injection therapy in multiple sclerosis treatment.

Herein, we evaluate the safety and general toxicology of GA inhalation in a porcine model of repeated GA inhalations by nebulizer administration. We assess airway peak pressure ($P_{peak}$) as an indicator of bronchoconstriction; and white blood cell counts, inflammatory parameters and pulmonary histopathology for detection of systemic or local inflammation.

## Materials and methods

### Experimental materials

**Glatiramer acetate, GA.** The clinically-used GA formulation (PubChem CID: 3,081,884; Copaxone®, Teva Pharmaceutical Industries Ltd., Petah Tikva, Israel) was obtained from Aarhus University Hospital pharmacy as pre-filled syringes containing 40 mg of GA.

**Study animals and groups.** Twelve young adult female outbreed Danish agricultural pigs were obtained from a local, conventional farmer and kept at Påskehøjgård, Aarhus, with unlimited access to food and water. The average body weight was 41 kg at the start of the study. The study protocol was approved by The Animal Experiments Inspectorate (Danish Veterinary and Food Administration, Ministry of Environment and Food of Denmark), ref. no. 014−15−0201−00441 in accordance with the Directive 2010/63/EU, European Parliament

on the protection of animals used for scientific purposes [11,12]. Four animals were used for pilot studies and excluded from the final results. The remaining eight pigs were divided into three groups: i) Four pigs inhaled 40 mg of nebulized GA (Lot C42137) diluted in isotonic saline to a total of 5 ml; ii) two pigs received a subcutaneous (*s.c.*), injection of 40 mg GA; iii) two pigs inhaled 40 mg of nebulized mannitol (inactive ingredient in Copaxone®) diluted in 5 ml of isotone saline. Mannitol was chosen as it is often used as an inhaled agent in patients with cystic fibrosis to loosen mucus in the airways [13]. Each animal received inhaled GA, mannitol or GA *s.c.* three times with two to three weeks interval before being euthanized.

## Experimental procedures

**Sedation and intubation.** Premedication was given with 125 mg tiletamine and 125 mg zolazepam (Zoletil mix., Virbac, Taguig City, Philippines) and after 10 minutes intravenous access was established in an ear vein and the pig was intubated and the ventilation controlled by a ventilator (GE Datex-Ohmeda Advance S5 Anaesthesia Machine, Soma Technology, Bloomfield, USA). Anaesthesia was maintained with a continuous intravenous infusion of Propofol (B. Braun Medical A/S, Copenhagen, Denmark). Following intubation and anaesthesia induction, the animals where observed for at least 30 minutes prior to the intervention, which was initiated at a time point where they were fully stable regarding haemodynamics and ventilation. At the end of each experiment, the anaesthesia was stopped and when the swallowing reflex had returned the pigs were extubated and transported to the stable. After the last experimental round, the animals were not woken, but instead euthanized with an intracardiac injection of pentobarbital while still in general anaesthesia.

**GA administration.** 40 mg GA (equivalent to 36 mg glatiramer base in one 1 ml vial) GA were diluted to a final volume of 5 ml in isotonic saline. The solution was then dispensed in a Jet-nebulizer (CirrusTM 2, INTERSURGICAL nebulizer universal Mouthpiece T-kit, Wokingham, UK) inserted between the tracheal tube and the ventilator tubing. The intratracheal route was chosen as the animals had to be in general anaesthesia connected to a ventilator during recording of pulmonary parameters. The GA was administered over ten minutes via nebulization with an airflow of 8 l/min. Approximately, 74% of the volume output will be particles less than five microns in diameter with a mass median diameter (MMD) of 3.3 microns.

**Ventilator settings and pulmonary parameters.** Bronchoconstriction was evaluated mainly by ventilator $P_{peak}$ and subsequently by clinical exam as wheezing and prolonged expiration. $P_{peak}$ (cm $H_2O$) and $pO_2$ were measured before, during and after the administration of nebulized GA or mannitol or GA *s.c.* $P_{peak}$ can be measured when the ventilator is set on a volume-controlled mode with a constant inspiratory flow. In this mode, any bronchoconstriction resulting in increased airway resistance will result in increased $P_{peak}$:

Resistance = Δ pressure / flow

The $P_{peak}$ was recorded from the ventilator and the $pO_2$ recorded from a transcutaneous probe both every five minutes. The pulse was recorded every five minutes and the electrocardiogram followed on a monitor.

**Blood sampling.** For flow cytometry, blood samples were collected aseptically by venepuncture from the jugular vein into a sterile blood collection tube ($K_3$ EDTA Vacutainer tubes, Becton Dickinson, Franklin Lakes, NJ). For biochemical analyses, all samples were collected in EDTA tubes (Becton Dickinson). Blood samples were drawn immediately before and after intervention.

**Biochemical analyses.** Plasma creatinine, ALAT, ASAT and CRP were measured by photometry (Cobas 6000, Roche Diagnostics, Hvidovre, Denmark) in each animal during anaesthesia before and after administration of GA inhalation, GA *s.c.* or mannitol inhalation. The

analyses were conducted by an experienced laboratory technician at the Department of Clinical Biochemistry, Aarhus University Hospital, Aarhus, Denmark.

**Immune cell analyses by flow cytometry.** Direct immunofluorescence staining of 50 μl whole blood was conducted by addition of 2.5 μl of anti CD14-FITC (Bio-Rad Laboratories, Inc., Hercules, CA), 10 μl of anti CD18-APC (Becton Dickinson), 1 μl of anti CD5-FITC (Bio-Rad), 20 μl of anti CD79a-PE (Bio-Rad), 1 μl of anti CD3-FITC (Bio-Rad), 1 μl of anti CD8a-AF647 (Becton Dickinson), 1 μl of anti CD4a-PE-Cy7 (Becton Dickinson), 1 μl of anti CD16-PE (Bio-Rad) and 1 μl of anti CD172a-FITC (Becton Dickinson). Samples were vortexed gently and incubated at room temperature for 15 min protected from light. Stained samples were treated with 425 μl FACS lysing solution (Becton Dickinson) to lyse erythrocytes while preserving the leukocytes. Samples were briefly vortexed and incubated in the dark for 20 min at room temperature. Cells were washed twice in PBS supplemented with 0.5% BSA (w/v) and 0.09% $NaN_3$ (v/v), centrifuged at 230xg for 5 min and finally resuspended in 250 μl PBS supplemented with 0.5% BSA (w/v) and 0.09% $NaN_3$ (v/v). Flow cytometry data were collected on a Novocyte™ (ACEA Biosciences Inc., San Diego, CA) and analysed using FlowJo V. 10.0.8 software (FlowJo, Inc., Ashland, OR). The Novocyte™ enables accurate volumetric-based cell counting for absolute cell counts without the need of reference counting beads.

**Histopathology.** Immediately after euthanizing, the lungs were removed and sections from the main bronchi and peripheral parts including small bronchioles were preserved in neutral-buffered, 10% formalin solution, blinded and stored for one week. The samples were subsequently processed, embedded in paraffin, sectioned at approximately 5 μm and stained with haematoxylin and eosin (H&E). Microscopic examination included evaluation of the main bronchi (carina) and peripheral lung tissue for the presence or absence of significant tissue alterations and signs of inflammatory infiltrate.

**Statistics.** $P_{peak}$ was modelled using a regression with random intercepts for each pig and each day, with independent residuals, and with a variance component for each day. Model control showed an acceptable model fit. Margins and marginal effects were estimated.

A "Two-one-sided t-test" (TOST) [14], was used to test equivalence in effects between GA inhaled and both GA *s.c.* and mannitol with respect to a measurement error on $P_{peak}$ of 2 $cmH_2O$ (Datex-Ohmeda Avance—User Reference Manual).

Comparison of blood cell counts was done based on regression similar to $P_{peak}$. The effects (POST—PRE) were compared as differences with confidence intervals between mannitol and GA inhaled and between GA *s.c.* and GA inhaled in Forest plot.

## Results

The three administrations were done with an interval of two weeks between the first and second and the third three weeks after the second administration. Over the five weeks study period all pigs gained weight; the mannitol group 7.6 kg, the GA *s.c.* group 7.0 kg and the GA inhalation group 9.8 kg showing that overall the pigs did well and were in good health. None of the measured vital parameters varied significantly between the three sessions or between the three groups (vital record data available here: DOI 10.6084/m9.figshare.9003164). A local inflammatory response was observed at the injection site circumference (approx. 5 cm ø) on the two pigs receiving GA *s.c.*, but this is a known side effect (> 10%) [15]. Apart from this, no clinically abnormal signs were detected.

### GA inhalation did not cause bronchoconstriction

Measured by a transcutaneous probe the $pO_2$ remained stable in all animals during all sessions between 98% and 100% and was not analysed further. None of the animals showed clinical

signs of bronchoconstriction observed as wheezing or prolonged expiration. The primary measure of interest was $P_{peak}$ recorded by the ventilator, as $P_{peak}$ is closely related to any bronchoconstriction that may occur during inhalation. A total of five hundred data points was available for $P_{peak}$ statistical analyses. $P_{peak}$ remained low during the entire experiment, with an insignificant increase in all three groups during anaesthesia (Table 1), which could be expected as lung compliance are known to decrease slightly during anaesthesia in general. The effect of GA inhalation on $P_{peak}$ was tested equivalent (max P-value = 0.00) to the effects of both GA *s.c.* and mannitol inhalation with respect to the ventilator measurement error on $P_{peak}$ of ± 2 cmH$_2$O. The TOST tests are visually confirmed by Fig 1 (all statistical analyses performed can be accessed here: DOI 10.6084/m9.figshare.9003164). Our results indicate that GA inhalation does not cause bronchoconstriction.

## Biochemistry and bronchial mucosa were normal after GA inhalation

Routine photometry of blood samples showed that all CRP values measured remained below detection limit ($< 0.6$ mg dl$^{-1}$) throughout the study. Likewise, all ALAT, ASAT and p-creatinine measurements were within the normal range and no significant changes were detected (data available here: DOI 10.6084/m9.figshare.9003164). Histopathology of the bronchi and bronchioles showed normal bronchial mucosa with ciliated pseudostratified columnar epithelium with goblet cells in all samples from pigs receiving GA inhalation, GA *s.c.* and mannitol inhalation. There were no signs of inflammatory infiltrate in either the mucosa or submucosa, and the mucus lining the luminal surface was acellular (data summary available in a tabular form here: DOI 10.6084/m9.figshare.9003164).

## GA inhalation affects white blood cell subsets similar to GA s.c.

As GA has an immunomodulating effect when administered *s.c.*, we studied if inhaled GA influenced white blood cell subsets. This was done by immune cell analyses by flow cytometry. Except for CD16+ cells favouring GA inhalation, no difference in effect was detected for the GA inhalation compared with the GA *s.c.* group ($p < 0.05$, Fig 2). Compared with mannitol inhalation, the effect of GA inhalation was significantly larger for monocytes, CD3+ and CD4a + cells (Fig 2), but smaller for NK and CD16+ cells ($p < 0.05$, Fig 2). The CD3+ increase found in both GA groups indicates that T-cells are GA sensitive, which is consistent with our previous findings [9]. We cannot thoroughly explain the mechanisms that lies behind this, but GA is known to exhibit immunomodulatory effects [16].

**Table 1. GA inhalation effect on bronchoconstriction measured as peak pressure, $P_{peak}$.**

| Group | Time | Mean (95% CI) | Effect (95% CI) |
|---|---|---|---|
| **Mannitol** | Pre | 19.41 [18.18; 20.65] | |
| | Post | 19.94 [18.78; 21.10] | 0.523 [[0.050; 0.997] |
| **GA *s.c.*** | Pre | 20.40 [19.14; 21.65] | |
| | Post | 21.18 [20.02; 22.34] | 0.783 [0.263; 1.303] |
| **GA inhaled** | Pre | 19.32 [18.47; 20.18] | |
| | Post | 20.53 [19.71; 21.35] | 1.213 [0.942; 1.483] |

Estimated means and effect with 95% CI of $P_{peak}$ in cm H$_2$O pre and post inhalation for pigs receiving mannitol inhalation, GA s.c. or GA inhalation. All changes are within the pressure monitoring accuracy (± 2 cm H$_2$O), and thus not biologically different.

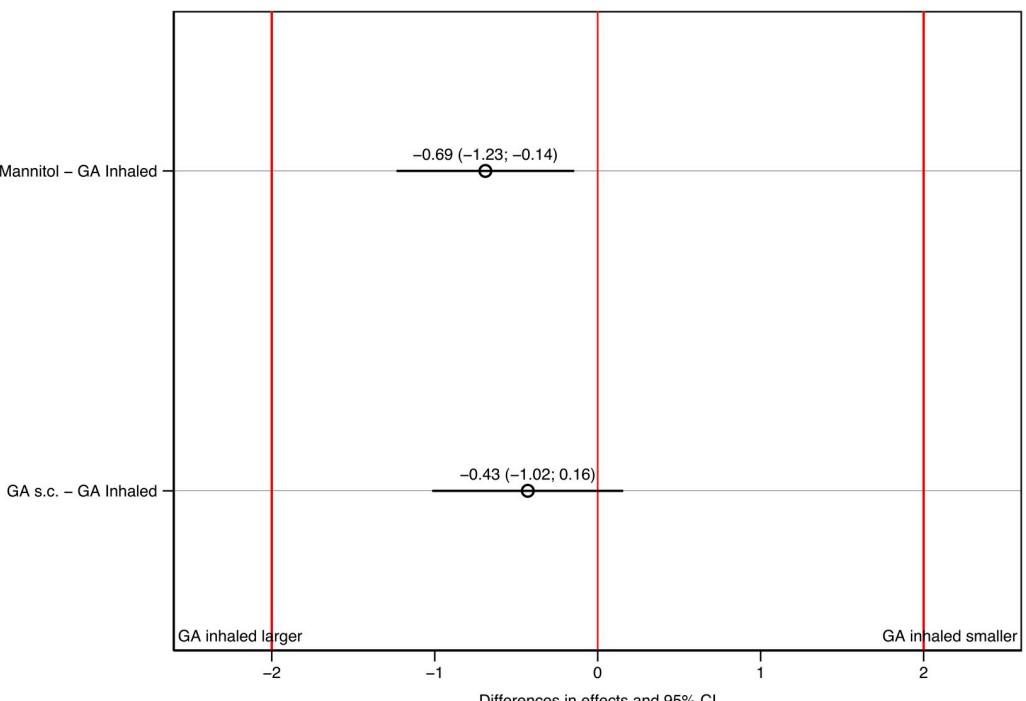

**Fig 1. Comparing the effect on $P_{peak}$ of GA inhaled with GA s.c. and mannitol.** Although there is a statistically significant larger effect for GA inhaled with respect to the effect of mannitol (confidence interval above red line at 0), both confidence intervals lay within the bounds of the measurement error (-2 and 2 $cmH_2O$), red vertical lines), i.e. they are equivalent and clinically insignificant.

## Discussion

Pulmonary infections with multidrug-resistant bacteria are life-threatening and current treatment options are not sufficient, why new approaches are necessary. We suggest that GA is repurposed for inhalation therapy that can be included in treatment strategies in combination with conventional antibiotics. Thus, we investigated the safety of inhaling GA in a pig model. We found that our animals tolerated GA inhalation well and that there were no signs of bronchoconstriction assessed by increased $P_{peak}$ or decreased $pO_2$, as well as no signs of inhalation-induced airway inflammation.

The usual systemically administered doses used in adult humans are 20 mg GA *s.c.* once-daily or 40 mg *s.c.* three-times-a-week. Our animals inhaled a double-daily human dose. This was estimated based on the average weight of an adult of 70 kg compared to our pig's average body weight of 41 kg. Thus, we consider the 40 mg administered directly to the lungs a high dose. The GA concentration proven effective by Christiansen et al in killing *P. aeruginosa* planktonic cells and biofilms was 25 µg $ml^{-1}$ and 640 µg $ml^{-1}$ [4]. Thus, it is reasonable to assume that an effective concentration can be achieved by GA inhalation of e.g. 5 mg administered to the estimated 1–2.6 ml of epithelial lining fluid situated in bronchioles [17], depending on lung disease severity.

As GA is known to exhibit immunomodulatory effects, we searched for changes in different populations of immune cells. The GA mode of action in the treatment of multiple sclerosis is not fully elucidated [18], but it is beyond doubt that GA is modulating the immune response, which is also the case for several other antimicrobial peptides, including LL-37. GA has been found to have a multitude of effects on the immune system, including: i) binding to various

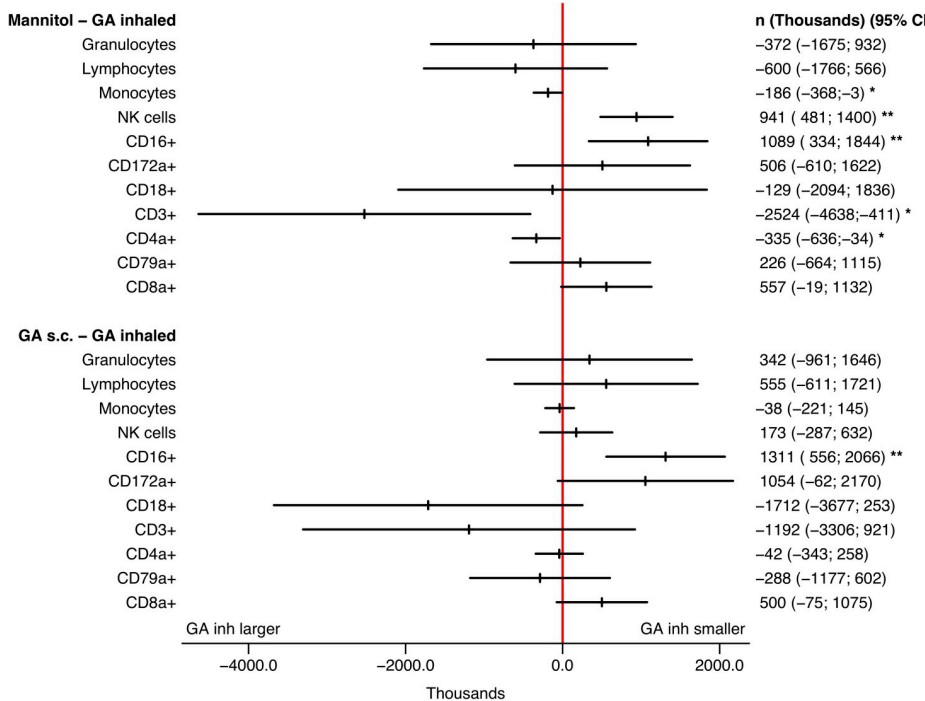

**Fig 2. WBC counts by flow cytometry (mean, 95% CI).** Change in effect (POST—PRE) in blood cell counts compared for mannitol and GA s.c. vs. GA inhaled. Effect (POST—PRE) is the increase in blood cell count for mannitol, GA s.c., and GA inhaled. The effects are compared as differences between mannitol and GA inhaled, as well as differences between GA s.c. and GA inhaled. The red line at zero should be within the shown confidence intervals if there are no difference between mannitol and GA inhaled (above); or GA s.c. and GA inhaled (below). * p < 0.05; ** p < 0.01.

MHC class II molecules; ii) inhibitory effect on monocytes reactivity; iii) deviation of dendritic cells and monocytes to produce less TNF-α and IL-12, more IL-10 and TGF-β; iv) to stimulate Th2 anti-inflammatory responses by induction of specific Th2/3-cells that secrete high amounts of IL-4, IL-5, IL-10, and TGF-β; v) prevalence and function elevation of T-regulatory cells; vi) activation of the transcription factor Foxp3; vii) reduction of Th-17 cells and their transcription factors RORγt; viii) improvement of the regulatory function of CD8+ T-cells; ix) reduction in the overall expression of IFN-γ; decrease in the amount of Th-17 cells and an increase in T-regulatory cells [16]. Here, no alterations in white blood cell counts were observed between the GA *s.c.* and the GA inhalation groups except for CD16+ cells, neither where there any local infiltrate of inflammatory cells detected by lower respiratory tract histopathology. A few alterations between the GA inhalation group and the mannitol group was evident, suggesting that some pulmonary GA uptake might occur. Yet for now, no data exists on the proportion of inhaled GA that is absorbed systemically, why this should be assessed in a larger scale study prior to clinical implementation.

We acknowledge that including only females is a limitation; however, boars were excluded in this preliminary study as they are more aggressive and require bigger, individual stalls according to the regulations [11,12]. Nevertheless, within the modest number of animals included limited to females and three administration rounds, our data suggests that GA can safely be inhaled in doses that may exhibit substantial bactericidal effect *in situ*.

Antimicrobial peptides are a part of non-specific innate immunity in almost all life domains [19]. However, according to the existing literature, their successful application has likely been

hampered by toxicity issues [20]. This was the case for colistin, a naturally occurring antimicrobial peptide, which was previously abandoned due to toxicity. Nevertheless, colistin was recently reintroduced due to development of multidrug-resistance and is today considered as an antibiotic of last resort for infections caused by multidrug-resistant Gram-negatives such as *P. aeruginosa* and *A. baumannii* [20]. Known side effects of colistin inhalation include irritative respiratory tract symptoms, here among dyspnoea and bronchospasm. Thus, toxicity is a considerable concern for pulmonary application of antimicrobial peptides, but these may also be key in managing chronic lung infections developing multidrug-resistant strains due to repeated antibiotic exposure, as well as nosocomial pulmonary infections caused by multidrug-resistant strains.

Currently, approximately twenty antimicrobial peptides are investigated in preclinical or clinical trials, only few already available in the clinic and mainly for topical application [10]. GA has been used for more than twenty years, it is generally safe, and no serious adverse events have been recorded [6,7]. Furthermore, GA is the preferred drug of choice for pregnant multiple sclerosis patients. Thus, repurposing GA as an antimicrobial against lethal infections is a promising strategy that can quickly be implemented in clinical settings and will likely not be compromised by the known safety profile [4]. Advantageously, GA can be combined with other antimicrobials as other antimicrobial peptides have been demonstrated to enhance antibiotic activity, lowering the antibiotic concentrations needed [21]. Thus, a synergistic effect would be expected, owing to the distinct mechanism of action resulting in increased membrane permeability. This could possibly re-introduce the use of antibiotics that are otherwise abandoned due to resistance and poor biofilm activity. For instance, GA could be combined with antibiotics where the resistance mechanism involves the bacterial membrane or where the permeability is poor [22,23]. Furthermore, GA could advantageously be implemented in the treatment of co-colonized patients, as it exhibits activity towards several relevant Gram-negatives and to some extent also Gram-positives [4]. A broad-spectrum range, strong synergy with conventional antibiotics and low probability of resistance development characterize antimicrobial peptides in general [21].

In conclusion, our study suggests that inhalation of GA is well-tolerated and safe, as it did not induce neither bronchoconstriction, nor changes in vital parameters, nor inflammation in the lower respiratory tract with the high GA dose used. We did not observe any potential side effects of inhalant GA, but the potential immunomodulatory effects would have to be more thoroughly addressed in a larger scale study prior to clinical implementation. Nevertheless, a new use of an already approved drug is a shortcut to meet the challenges in our combat against pathogens. This preliminary study may pave the way for clinical trials in patients colonized in their airways with multidrug resistant bacteria and other patients with chronic lung infections caused by Gram-negatives such as *P. aeruginosa*.

## Acknowledgments

The authors would like to thank Kira Sonnichsen Graahede (veterinary technician), Bettina Winther Grumsen (laboratory technician) and Tine Vasegaard (medical laboratory technician) for excellent technical assistance.

## Author Contributions

**Conceptualization:** Thomas Vorup-Jensen, Eskild Petersen.

**Data curation:** Sandra M. Skovdal, Stig Hill Christiansen, Karen Singers Johansen, Eskild Petersen.

**Formal analysis:** Sandra M. Skovdal, Stig Hill Christiansen, Niels Henrik Bruun, Eskild Petersen.

**Funding acquisition:** Stig Hill Christiansen, Thomas Vorup-Jensen, Eskild Petersen.

**Investigation:** Sandra M. Skovdal, Stig Hill Christiansen, Karen Singers Johansen, Ida Elisabeth Holm, Eskild Petersen.

**Methodology:** Stig Hill Christiansen, Karen Singers Johansen, Ole Viborg, Thomas Vorup-Jensen, Eskild Petersen.

**Project administration:** Stig Hill Christiansen, Thomas Vorup-Jensen, Eskild Petersen.

**Resources:** Ole Viborg, Thomas Vorup-Jensen, Eskild Petersen.

**Software:** Niels Henrik Bruun.

**Supervision:** Ole Viborg, Søren Jensen-Fangel, Thomas Vorup-Jensen, Eskild Petersen.

**Validation:** Ole Viborg, Thomas Vorup-Jensen, Eskild Petersen.

**Visualization:** Sandra M. Skovdal.

**Writing – original draft:** Sandra M. Skovdal, Eskild Petersen.

**Writing – review & editing:** Sandra M. Skovdal, Stig Hill Christiansen, Niels Henrik Bruun, Søren Jensen-Fangel, Ida Elisabeth Holm, Thomas Vorup-Jensen, Eskild Petersen.

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
