## [Decision Letter · Decision Letter 0]

18 Jul 2019

PONE-D-19-17175

Inhaled nebulized glatiramer acetate against Gram-negative bacteria does not induce pulmonary side effects in adult pigs

PLOS ONE

Dear Prof. Vorup-Jensen,

Thank you for submitting your manuscript to PLOS ONE. After careful consideration, we feel that it has merit but does not fully meet PLOS ONE’s publication criteria as it currently stands. Therefore, we invite you to submit a revised version of the manuscript that addresses the points raised during the review process.

We would appreciate receiving your revised manuscript by Sep 01 2019 11:59PM. To enhance the reproducibility of your results, we recommend that if applicable you deposit your laboratory protocols in protocols.io, where a protocol can be assigned its own identifier (DOI) such that it can be cited independently in the future. For instructions see: http://journals.plos.org/plosone/s/submission-guidelines#loc-laboratory-protocols

We look forward to receiving your revised manuscript.

Kind regards,

Monica Cartelle Gestal, PhD

Academic Editor

PLOS ONE

Journal Requirements:

3. Please note that all PLOS journals ask authors to adhere to our policies for sharing of data and materials: https://journals.plos.org/plosone/s/data-availability. According to PLOS ONE’s Data Availability policy, we require that the minimal dataset underlying results reported in the submission must be made immediately and freely available at the time of publication. As such, please remove any instances of 'unpublished data' or 'data not shown' in your manuscript and replace these with either the relevant data (in the form of additional figures, tables or descriptive text, as appropriate), a citation to where the data can be found, or remove altogether any statements supported by data not presented in the manuscript."

4. We note that you have a patent relating to material pertinent to this article. Please provide an amended statement of Competing Interests to declare this patent (with details including name and number), along with any other relevant declarations relating to employment, consultancy, patents, products in development or modified products etc. Please confirm that this does not alter your adherence to all PLOS ONE policies on sharing data and materials, as detailed online in our guide for authors http://journals.plos.org/plosone/s/competing-interests by including the following statement: "This does not alter our adherence to  PLOS ONE policies on sharing data and materials.” If there are restrictions on sharing of data and/or materials, please state these. Please note that we cannot proceed with consideration of your article until this information has been declared.

Additional Editor Comments:

Please address all the comments of the reviewers

Reviewers' comments:

Reviewer's Responses to Questions

**Comments to the Author**

1. Is the manuscript technically sound, and do the data support the conclusions?

Reviewer #1: Partly

Reviewer #2: Yes

2. Has the statistical analysis been performed appropriately and rigorously? 

Reviewer #1: Yes

Reviewer #2: I Don't Know

3. Have the authors made all data underlying the findings in their manuscript fully available?

Reviewer #1: Yes

Reviewer #2: Yes

4. Is the manuscript presented in an intelligible fashion and written in standard English?

Reviewer #1: Yes

Reviewer #2: Yes

5. Review Comments to the Author

Reviewer #1: The current study looked at the safety of glatiramer acetate (GA) intratracheal nebulization in healthy, young adult female pigs. While the experiments, statistics, and most analyses were performed to a high technical standard and described in sufficient detail, the reviewer has some concerns that may affect the conclusions presented. Additional comments are included in the revised manuscript.

1. In general safety studies includes both sexes. The current study only looked at female pigs. Why were males not included? Please explain and justify.

2. Barbiturates tend to relax smooth muscle during anesthesia. Could the use of propofol have influenced the measurement of Ppeak?

3. In a clinical setting, administration of GA would have been given by oro-nasal route; right? In the current study, this would have changed the assessment of safety parameters, including, at the very least, histopathological evaluation of the upper respiratory tract. Why was the intratracheal route chosen over the oro-nasal route? Please explain and justify.

4. Why not do flow cytometry in lung tissue and look for the various cell types (monocytes, T-cells, and others), including eosinophils and plasma cells, which play an important role in the generation of an immediate and delayed reaction that can lead to broncho- and bronchiolo-constriction? See reference: Yu Y-RA, O’Koren EG, Hotten DF, Kan MJ, Kopin D, Nelson ER, et al. (2016) A Protocol for the Comprehensive Flow Cytometric Analysis of Immune Cells in Normal and Inflamed Murine Non-Lymphoid Tissues. PLoS ONE 11(3): e0150606. doi:10.1371/ journal.pone.0150606

5. Who served as the study pathologist for the study?

a. What were his/hers credentials?

i. European College of Veterinary Pathology?

ii. American College of Veterinary Pathology?

b. Was the study read blindly (masked) at least initially?

c. Were there any fresh samples collected for analysis? If so, what analysis, if any?

i. This would have been a good opportunity to do microbial isolation, flow cytometry, PCR, etc.

d. Was the lungs infused with fixative?

e. What did the histopathological evaluation consisted of? What were the parameters observed? Their incidence? Their severity? Please be specific.

Example of how I would have approached this:

i. “Microscopic exam consisted of evaluation of the trachea, bronchi, bronchioles, respiratory bronchioles, and alveolar sacs from all lung lobes (the pig has 7 lobes) for the presence or absence of significant tissue alterations (lesions), including but not limited to inflammation, degeneration, and necrosis. Microscopically, lesion (tissue alteration) incidence, severity, and distribution were recorded. Histopathologic severity scores were assigned as grades 0 (no significant histopathological alterations); 1 (minimal); 2 (mild); 3 (moderate); or 4 (severe) based on an increasing extent and/or complexity of change, unless otherwise specified. Lesion distribution was recorded as focal, multifocal, or diffuse, with distribution scores of 1, 2, or 3, respectively.”

In addition, the histopathology data needs to be presented in a tabular format.

Example:

ii. “The incidence, severity, and distribution of significant microscopic findings are summarized on the Project Tabulated Animal Data - Summary - Microscopic Findings, in which the numbers of animals per group and lesions per group are indicated. Project Tabulated Animal Data – Individual Animals – Microscopic Findings present the microscopic findings for individual animals by group. The codes and abbreviations used as entries in these tables are explained at the bottom of each table.”

iii. More information can be found at: Kenneth A. Schafer, John Eighmy, James D. Fikes, Wendy G. Halpern, Renee R. Hukkanen, Gerald G. Long, Emily K. Meseck, Daniel J. Patrick, Michael S. Thibodeau, Charles E. Wood, Sabine Francke. 2018. Use of Severity Grades to Characterize Histopathologic Changes. Toxicol Pathol 46, 256-265; https://doi.org/10.1177/0192623318761348

6. Did you do histopathology at the injection site? If not, then why not? Subcutaneous lipoatrophy, necrosis, and others have been associated with administration of GA.

7. Data from pig #18 had a wide range in Ppeak. Although this may not be clinically significant, this was a healthy pig nevertheless. I would expect the difference be more accentuated in sick animal. Is this interpretation correct?

8. Pigs coming directly from the farm, such as the ones from the current study, usually have some sort of bronchial-associated lymphoid hyperplasia, often secondary to Mycoplasma ssp. infections, which potentially makes the airways hypersensitive to stimuli, such as GA. Did you see any of that? Other potential lung infections from the farm include Bordetella, Haemophilus, Streptococcus, several viruses, and parasites. Were these pigs checked for non-clinical infections? Again, who did the evaluation of the lungs? What were his/hers credentials?

Reviewer #2: The manuscript “Inhaled nebulized glatiramer acetate against Gram-negative bacteria does not induce

pulmonary side effects in adult pigs” by Skovdal et al describes results of assessing the side effects on respiratory exposure of animals (pigs) with therapeutically ascertained doses of the random co-polymer glatiramer acetate (GA). The compound is currently in use as an FDA approved compound to treat MS patients by intravenous injections though observed to have anti-inflammatory and leukocytic activities. GA was subsequently see to possess antimicrobial activity similar to the cationic peptide LL-37 The manuscripts starts with the premise that the drug is inherently safe within the body.

Major Comments

I feel, in its present form, the narrow focus of the preliminary work, investigating the safety in the putative use of GA as a therapeutic may be more appropriate in journals focused on such studies. As such the quantum of work currently presented in the manuscript does not reach the threshold of an independent publication and would be more suited as preliminary work/data accompanying a broader investigation, namely the therapeutic benefit of GA in animal models of respiratory infection.

Therefore, this reviewer feels that in the current form the manuscript cannot be published and requires a major revision.

Minor comments on manuscript

1. There are few technical deficiencies in the manuscript. Though the control groups of animals used (n=2) are small in number and may be straining statistical results.

2. The language is sufficiently clear for the most part except for some clarity required in animal number assignments in the Materials ( line 95: 12 pigs; Line 101 “four pigs used for pilot studies” Line “8 pigs divided into 3 groups” )

3. The Results section begins without any compiled description of the experimental format (the descriptions of the experiments are described in bits and pieces among various parts of Materials section).

4. The manuscript may be better served if the background of the earlier studies (some conducted by the authors group) is elaborated in the introduction to increase the contextual significance of the manuscript.

6. PLOS authors have the option to publish the peer review history of their article (what does this mean?). If published, this will include your full peer review and any attached files.

Reviewer #1: Yes: Uriel Blas-Machado

Reviewer #2: No

---

## [Author Response · Author response to Decision Letter 0]

29 Aug 2019

(a file has also been submitted with our replies)

---

## [Decision Letter · Decision Letter 1]

26 Sep 2019

Inhaled nebulized glatiramer acetate against Gram-negative bacteria is not associated with adverse pulmonary reactions in healthy, young adult female pigs

PONE-D-19-17175R1

Dear Dr. Vorup-Jensen,

We are pleased to inform you that your manuscript has been judged scientifically suitable for publication and will be formally accepted for publication once it complies with all outstanding technical requirements.

With kind regards,

Monica Cartelle Gestal, PhD

Academic Editor

PLOS ONE

Additional Editor Comments (optional):

Dear Dr. Vorup-Jensen,

Now all the concerns have neen addresssed I am pleased to announce that the manuscript has been accepted for publication in our journal.

Reviewers' comments:

Reviewer's Responses to Questions

**Comments to the Author**

1. If the authors have adequately addressed your comments raised in a previous round of review and you feel that this manuscript is now acceptable for publication, you may indicate that here to bypass the “Comments to the Author” section, enter your conflict of interest statement in the “Confidential to Editor” section, and submit your "Accept" recommendation.

Reviewer #1: All comments have been addressed

Reviewer #2: All comments have been addressed

2. Is the manuscript technically sound, and do the data support the conclusions?

Reviewer #1: Yes

Reviewer #2: Yes

3. Has the statistical analysis been performed appropriately and rigorously? 

Reviewer #1: Yes

Reviewer #2: Yes

4. Have the authors made all data underlying the findings in their manuscript fully available?

Reviewer #1: Yes

Reviewer #2: Yes

5. Is the manuscript presented in an intelligible fashion and written in standard English?

Reviewer #1: Yes

Reviewer #2: Yes

6. Review Comments to the Author

Reviewer #1: (No Response)

Reviewer #2: (No Response)

7. PLOS authors have the option to publish the peer review history of their article (what does this mean?). If published, this will include your full peer review and any attached files.

Reviewer #1: Yes: Uriel Blas-Machado

Reviewer #2: No

---

## [Editor Report · Acceptance letter]

1 Oct 2019

PONE-D-19-17175R1 

Inhaled nebulized glatiramer acetate against Gram-negative bacteria is not associated with adverse pulmonary reactions in healthy, young adult female pigs 

Dear Dr. Vorup-Jensen:

I am pleased to inform you that your manuscript has been deemed suitable for publication in PLOS ONE. Congratulations! Your manuscript is now with our production department. 

With kind regards,

on behalf of

Dr. Monica Cartelle Gestal 

Academic Editor

PLOS ONE